# Design and Demonstration of an In-Plane Micro-Optical-Electro-Mechanical-System Accelerometer Based on Talbot Effect of Dual-Layer Gratings

**DOI:** 10.3390/mi14071301

**Published:** 2023-06-25

**Authors:** Wenqing Chen, Li Jin, Zhibin Wang, Haifeng Peng, Mengwei Li

**Affiliations:** 1School of Instrument and Electronics, North University of China, Taiyuan 030051, China; s202106096@st.nuc.edu.cn (W.C.); wangzhibin@nuc.edu.cn (Z.W.); sz202206104@st.nuc.edu.cn (H.P.); 2Academy for Advanced Interdisciplinary Research, North University of China, Taiyuan 030051, China

**Keywords:** optical micro-grating, Talbot effect, optical interference, micro-accelerometer

## Abstract

An ultrasensitive single-axis in-plane micro-optical-electro-mechanical-system (MOEMS) accelerometer based on the Talbot effect of dual-layer gratings is proposed. Based on the Talbot effect of gratings, the acceleration can be converted into the variation of diffraction intensity, thus changing the voltage signal of photodetectors. We investigated and optimized the design of the mechanical structure; the resonant frequency of the accelerometer is 1878.9 Hz and the mechanical sensitivity is 0.14 μm/g. And the optical grating parameters have also optimized with a period of 4 μm and a grating interval of 10 μm. The experimental results demonstrated that the in-plane MOEMS accelerometer with an optimal design achieved an acceleration sensitivity of 0.74 V/g (with better than 0.4% nonlinearity), a bias stability of 75 μg and an acceleration resolution of 2.0 mg, suggesting its potential applications in smartphones, automotive electronics, and structural health detection.

## 1. Introduction

At present, MEMS accelerometers are widely used in the fields of smartphones, automotive electronics and seismic and structural health detection [1,2,3,4,5]. Different types of acceleration detection have been used, such as optical [6], capacitance [7], piezoresistive [8], piezoelectric [9] and so on [10,11]. Compared to other sensing technologies, capacitive sensing is highly attractive in the literature due to its ease of compatibility with integrated circuit [12]. However, it is sensitive to electromagnetic interference. It is inevitably subject to parasitic capacitance and fringe effects, which will affect sensitivity and accuracy. Therefore, optical sensing has been considered as one of the most promising candidates in the field of structural health detection and inertial navigation due to its advantages of high precision, corrosion resistance and intrinsic immunity to electromagnetic interference [13,14,15].

Different types of MOEMS accelerometers have been investigated and significant results have been achieved [16,17,18,19,20]. Optical-fiber-based accelerometers have commonly been used for optical detection, but the integration of optical fibers with micromechanical structures in small bulks is not easy [21,22]. Grating-interference accelerometers have received much attention due to their advantages in terms of high sensitivity, low cost and lower power consumption [23]. However, they restrict the integration of multi-axis accelerometers on the same substrate [24]. With the advantages of a tunable linear range and ease of integration, a grating-Talbot-effect-based accelerometer has been proposed. In 2022, our group demonstrated a MOEMS out-of-plane accelerometer based on the Talbot effect of dual-layer gratings with a sensitivity of 3.1 V/g and a bias stability of 0.15 mg [25], which verified the feasibility of a grating-Talbot-based accelerometer with low noise, low power consumption and high sensitivity.

In this work, we propose an in-plane MOEMS accelerometer based on the Talbot effect of dual-layer gratings. This accelerometer takes advantage of a periodical change in Talbot pattern with high acceleration-displacement sensitivity. Using this approach, individual lateral and vertical axis accelerometers based on the Talbot effect can be fabricated on the same die, which can reduce the orthogonality error and optimize the performance of each axis. Moreover, the linear working range of MOEMS accelerometers can be designed and provided in certain applications. Here, the mechanical sensing structure is meticulously designed and optimized at a full scale range of 200 g. The experimental results show that the MOEMS accelerometer has a high sensitivity of 0.74 V/g, a bias stability of 75 µg and an acceleration resolution of 2.0 mg.

## 2. Sensing Principle

The optical mechanism of the in-plane grating-Talbot-based accelerometer is schematically described in Figure 1. When the laser light (model MDL-Ⅲ-1550, 1550 nm wavelength, with a collimated diameter of 800 µm) propagates to grating Ⅰ, the light travels through it and arrives at grating II (the interval of gratings is *d*_0_), and is deposited on the proof mass. The light is directly transmitted by grating Ⅱ, and this forms the Talbot effect of near-field diffraction. We record the diffraction intensity beneath the dual-layer gratings. The accelerometer structure consists of proof mass, dual-layer gratings (with the side length of *a*), spring beams and anchors. When it senses the acceleration in the *X* direction, the proof mass will be removed from its initial position, resulting in bending and distortion in the four spring beams, and the proof mass will move horizontally relative to grating Ⅰ along with the lateral movement of grating II in the sense direction (under the condition of ignoring the influence of cross-axis sensitivity). The relative position variation in dual gratings will lead to an intensity difference of near-field diffraction. Thus, the change can be measured to obtain the acceleration, which is detected by a photodetector (PD).

Based on a dual-layer gratings model, a series of simulations was carried out using the finite-difference time-domain (FDTD) method. When the collimated laser irradiates the dual-layer gratings, it will form a self-generated image of the grating pattern in the Z-direction at periodic intervals, which is called the Talbot effect as shown in Figure 2a. The periodic repetition of the field distribution in the near field of the grating can be seen through the Talbot effect transmitted by the gratings [26,27,28,29]. According to the theory of near-field diffraction gratings [30], the self-generated image is repeated with a Talbot period of approximately $d_{t}=\;2d^{2}/\lambda$ [31] and the Talbot imaging region is within $D<\pi d_{t}(2N-3)$/4, where *d* is the spatial period of the grating, *λ* is the wavelength of the incident light and *N* is the number of cycles of the laser beam passing through the grating. To obtain the maximum diffraction efficiency, the position of the lower grating is located at the position of the Talbot image with a tolerance of ±0.5 µm. Therefore, the interval between the upper and lower gratings was chosen to be 10 μm. We also investigated the influence of the grating parameters on the diffraction intensity distribution by optimizing the parameters of dual gratings; the grating parameter selected were a period of 4 μm, an Al grating thickness of 300 nm, and a duty ratio of 0.5. Figure 2b shows the relationship between the diffracted light and the displacement of the lower grating in the in-plane direction, with the period of *d* and linearity coefficients of 0.999 (Figure 2c). From the linear fitting, the sensitivity of the optical diffraction effect is 20%/μm. Although the optimal grating period is 2 μm, we still chose the grating period of 4 μm for processing due to the accuracy of the lithography (MA6-IV, with an accuracy of less than 1 μm). In the next step, the optical sensitivity and resolution of the accelerometer will be greatly improved by employing the stepping lithography (with an accuracy of 0.1 μm).

## 3. The Design and Fabrication of MOEMS Accelerometer

To obtain a mechanical structure with high sensitivity in the in-plane direction and low cross-axis sensitivity [32], we simulated the structure and optimized the parameters of the accelerometer (including the width and length of the cantilever beam and the size of the proof mass), and analyzed resonance frequency, stress and optimized parameters in detail [33]; the optimal parameters are shown in Table 1. Due to cross-axis interference being inevitable, we assessed the effect of deformation on the sensing axis when the non-sensing axis was subjected to acceleration. According to the simulation, the response sensitivity of the accelerometer in a non-sensing direction is 0.42 nm/g (Y-axis) and 0.29 nm/g (Z-axis), respectively.

Another important parameter of a MOEMS accelerometer is the operating bandwidth, which depends on the characteristics of mechanical structure. In order to calculate this parameter, we analyzed the resonant frequency of the accelerometer, which depends on the stiffness and proof mass [34]. Figure 3 shows the mechanical mode of the proposed accelerometer. The stresses are concentrated in the beam structure. The mass in the sensing direction is 15.3 mg, and the elastic coefficient is *K* = 1343 N/m. When the structure withstands an acceleration of 200 g, it reaches the allowed stress of silicon. The first mode frequency of the accelerometer is 1878.9 Hz, and the second, third and fourth mode frequencies are 8453.6 Hz, 12,172 Hz and 12,235 Hz, respectively. The resonant frequencies of other modes are far from the working mode, avoiding the coupling of other modes with the working mode and effectively suppressing the cross-axis sensitivity induced by other different modes. The bandwidth of the accelerometer is 626.3 Hz.

In the FEA simulation, the sensor has a displacement sensitivity of 140 nm/g. Due to the photoelectric conversion efficiency of PD being 0.86 A/W at a wavelength of 1550 nm, the photoelectric sensitivity of the grating-Talbot-based accelerometer can be calculated as 15.4 V/μm. Therefore, the total sensitivity of the MOEMS accelerometer can be calculated as 2.1 V/g.

The detailed fabrication of the MOEMS accelerometer using micro-nano technology is shown in Figure 4. Highly reflective films composed of aluminum are first patterned on the glass substrate using magnetron sputtering (Figure 4a). Then, the upper grating layer is fabricated by dry etching after spin coating a 1.5 μm thick photoresist (AZ5214) for photolithography (Figure 4b). To ensure the interval of the dual-layer grating, we etched a 10 μm shallow groove in the center of the silicon substrate (Figure 4c). For the subsequent electrostatic actuator and anodic bonding, the electrode grooves were etched by reactive ion etching (RIE) (Figure 4d). A 300 nm thick SiO_2_ layer was grown by PECVD to prevent conductivity of the electrodes to the wafer (Figure 4e). Though magnetron sputtering, spin-coating of a photoresist and photolithography, the lower grating and Al wires were also fabricated by metal dry etching (Figure 4f,g). The cantilever beams were also etched with deep reactive ion etching (DRIE) after wet etching the SiO_2_ layer (Figure 4h,i) and then releasing the sensitive structure (Figure 4j). Finally, the micro-grating layer and the structure layer were combined together via anodic bonding (Figure 4k).

## 4. Experiment and Discussion

A block diagram of the experimental setup for the static acceleration measurement test of the proposed accelerometer is shown in Figure 5. All of these configurations were mounted on a high-precision rotary table (model. AP180/M, Thorlabs, Shanghai, China) that generates input accelerations by rotating the table from 0° to 90°, which means that the input accelerations can be applied between 0 g and 1 g. A frequency-stabilized laser with a power of 1.1 mW served as the light source in our configuration, and the transresistance resistor of PD is 1.8 kΩ. To reduce the impact of frequency noise on device performance, we used phase-modulation techniques to suppress 1/*f* noise and other ambient noise. As shown in Figure 5, the realization of phase modulation requires a piezoelectric ceramic transducer (PZT), signal generator, demodulation module, low-pass filter and processing circuit. The signal generator sends two identical signals to a PZT and demodulation module, respectively. The diffraction intensity detected by the PD is modulated by applying a signal voltage to the PZT, then demodulated by multiplying the same frequency sinusoidal signal and loaded on low-pass filter and A/D converter to be recorded in the computer.

In order to measure the sensitivity of the accelerometer, we recorded experimental data at different values of acceleration. The average value was measured four times forward and backwards. A linear fitting for the output curve versus the acceleration along the sensitive axis is given in Figure 6a. The output of the MOEMS accelerometer can be written as follows:(1)$$V_{out}=0.74491x+9.00977$$
where *x* is the acceleration along the sense axis of the accelerometer. The sensitivity of the sensor is 0.745 V/g, and the *R*^2^ after linear fitting is 99.6%. From Figure 6a, it can be seen that the error bars of the measurement exist during the acceleration increasing and decreasing process, which may be derived from the inconsistency of the stiffness of the cantilever beams and thus different non-linearity forwards and backwards. In addition, the accuracy of the rotary table has a tolerance of ±0.3° during the measurement. Moreover, the direction of the non-sensitive axis is subject to acceleration provided by gravity, and the cross-axis sensitivity may influence the measurement accuracy. Figure 6b depicts the output signal of the root mean square (*RMS*) deviations when the applied acceleration on the accelerometer sensing axis is constant, which indicates that the noise level of the MOEMS accelerometer is 1.5 mV. Combined with acceleration sensitivity, the maximum acceleration resolution can be calculated according to the formula
(2)$$\mathrm{resolution}=\frac{\mathrm{Noise}\;\mathrm{density}}{\mathrm{Sensitivity}}=\frac{1.5\;\mathrm{mV}}{0.74\;V/g}=2.01\;\mathrm{mg}$$

In addition, long-term experimental data were recorded when the applied acceleration was zero. Figure 6c shows the Allan deviation of the raw data, in which the raw data have a sampling rate of 50 Hz. It demonstrates that the bias stability is 75 μg. For a better analysis of bias stability, we tested at different times of the day and the experimental results are shown in Figure 6d. It is clear that the fluctuations in the bias stability at different times of day is caused by fluctuations in temperature and environment vibrations, which affect the power of the laser and the stability of the accelerometer, thus contributing to the higher bias stability. The device’s performance can be further improved by reducing the laser intensity fluctuations and relative intensity noise and spectral purity of the laser.

## 5. Conclusions

In this paper, a MOEMS accelerometer based on the Talbot diffraction effect of dual-layer gratings is proposed. We simulated and analyzed the accelerometer working in a mode frequency of 1878.9 Hz and an operating bandwidth of 626.3 Hz. The experimental results demonstrated a sensitivity of 0.74 V/g with a better than 0.4% nonlinearity. Moreover, the Allan deviation suggests that the bias stability of the accelerometer is 75 μg, and the performance of the accelerometer is susceptible to environmental factors (such as fluctuations in temperature and environment vibrations). This research shows the feasibility of implementing an in-plane Talbot-effect-based MOEMS accelerometer. Further analysis is performed for the bias temperature sensitivity and three-axis integration on the same substrate.

## Figures and Tables

**Figure 1 micromachines-14-01301-f001:**
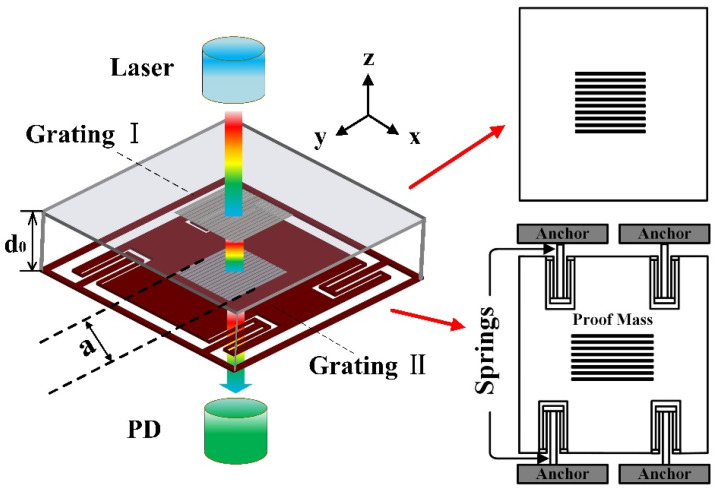
Schematic diagram of the dual-layer diffraction gratings-based accelerometer.

**Figure 2 micromachines-14-01301-f002:**
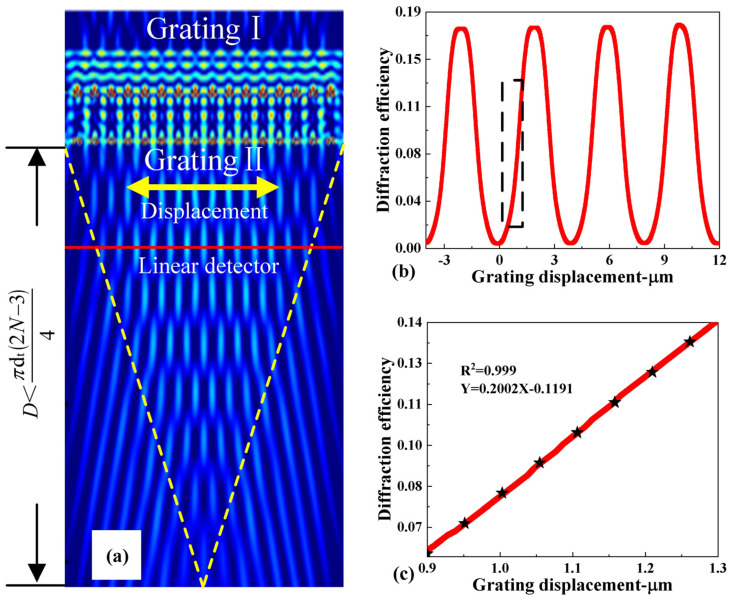
(**a**) The Talbot pattern of dual-layer diffraction gratings. (**b**) Simulation curve for the diffraction efficiency versus the displacement along the sensitive axis of accelerometer. (**c**) Linear fitting for the diffraction efficiency curve in the linear region of (**b**).

**Figure 3 micromachines-14-01301-f003:**
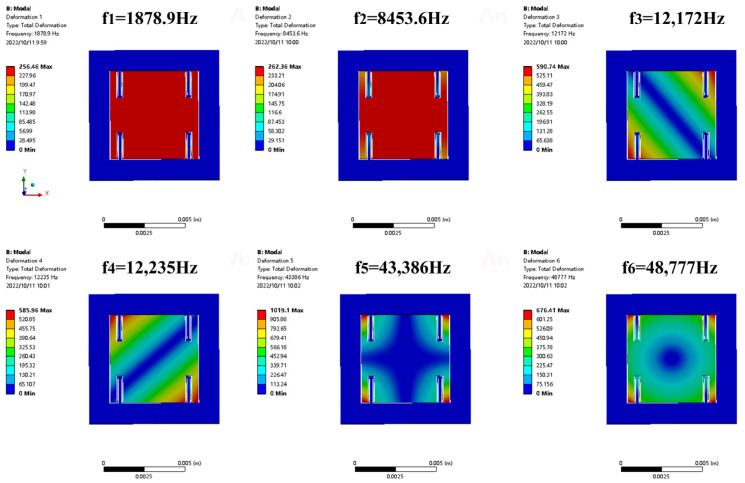
Simulation results of modal frequency of MOEMS accelerometer.

**Figure 4 micromachines-14-01301-f004:**
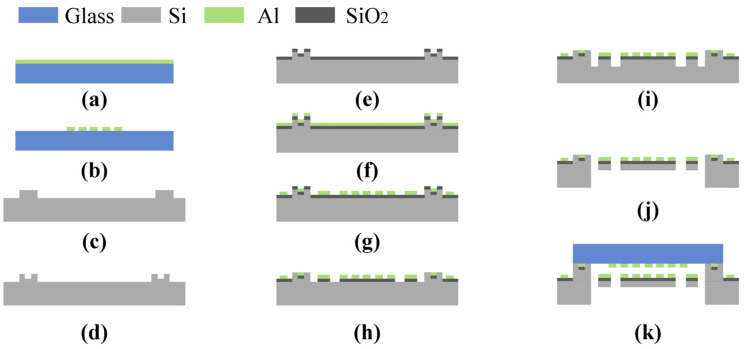
Fabrication process flow of the MOEMS accelerometer.

**Figure 5 micromachines-14-01301-f005:**
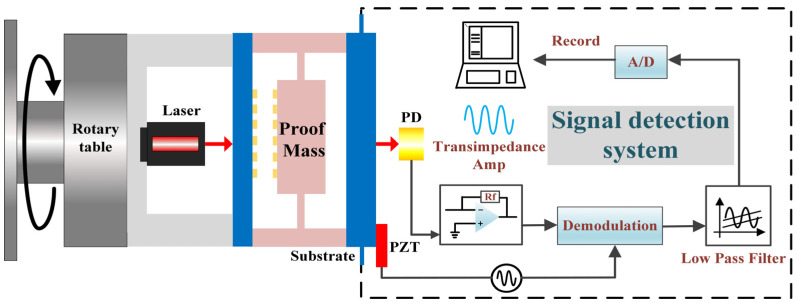
Experimental configuration for the static acceleration measurement.

**Figure 6 micromachines-14-01301-f006:**
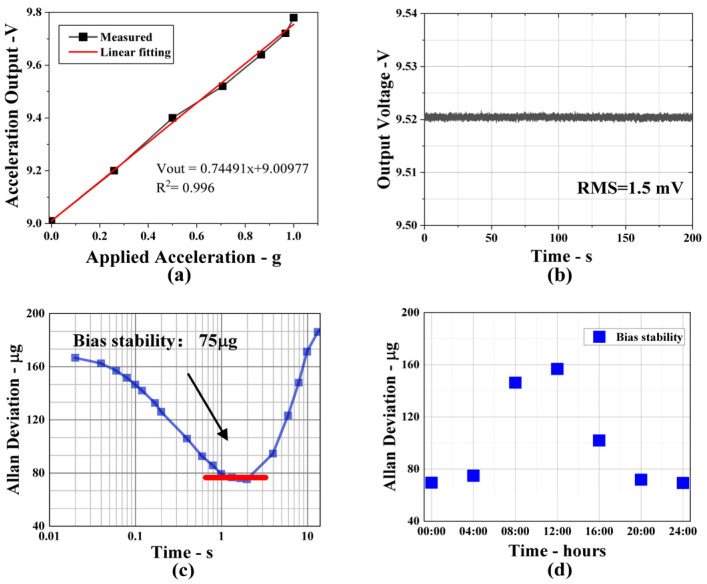
(**a**) Measured scale factor of the MOEMS accelerometer is 0.74 V/g. (**b**) The output voltage and RMS deviations of the accelerometer when the applied acceleration is invariable. (**c**) Modified Allan deviation of accelerometer. (**d**) Bias stability recorded at different times of the day.

**Table 1 micromachines-14-01301-t001:** The parameters of the accelerometer structure.

Design Parameter	Values
Young’s modulus of silicon	170 GPa
The Poisson ratio of silicon	0.28
Density of silicon	2329 kg/m^3^
Supporting beam length	1500 μm
Supporting beam width	150 μm
Sensing beam length	1300 μm
Sensing beam width	30 μm
The gap between two beams	50 μm
Frame length	4800 μm
Mass block thickness	100 μm

## Data Availability

The data that support the findings of this study are available from the corresponding author upon reasonable request.

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
