# Peer review of "Design and Demonstration of an In-Plane Micro-Optical-Electro-Mechanical-System Accelerometer Based on Talbot Effect of Dual-Layer Gratings"

_micromachines, 2023, doi:10.3390/mi14071301_

Round 1
Reviewer 1 Report
The paper presents the design and the demonstration of an in-plane MOEMS accelerometer based on Talbot effect using dual-layer gratings. This work is a follow-up of the same group research on out-of-plane accelerometer. Modeling of the different resonance modes is presented but the discussion of the in-plane versus out-of-plane responses is not done.
Linking and analyzing these behaviors will be valuable for the reader.
Comments are listed below.
1- Introduction
- Line 32: “high precision” instead of “highly precision”
2- Sensing principle
- Lines 65-66: the proof mass is not moving only in the horizontal plane, but also out-of-plane. The possible contribution/noise in the useful signal must be presented in the principle section, analysed in the modelling section and compared in the characterization section.
- Resolution of Figure 1 must be improved
- Line 82: “the lower grating is located EXACTLY at the position of the Talbot image”
What is the tolerance in the distance between the gratings? Explain the effect on the signal.
- Line 89-90: The sensitivity is “20%/µm”. But the maximal grating displacement is only 0.4µm, leading to a few % only for the sensitivity. The diffraction efficiency is low.
3- Design and fabrication
- Figure 3: The accelerometer exhibits different modes leading to the deformation of the grating surface, and then an effect on the wavefront, including shift, piston, tilt, astigmatism, focus … comment the magnitude of the effect and the impact on the performance (in the text)
4- Experiment
- Figure 6: what is the error bar of the measurement / measurement accuracy? Show/display in the figure, explain in the text
- Line 163: “MOEMS” instead of “MOMES”
- what is the efficiency of the device? Limited by the concept/architecture? Could be improved? Explain and comment.
General comment: Review English of the manuscript.
Conclusion of the review: Minor changes are mandatory before publishing.
Moderate editing of English language required.
Reviewer 2 Report
This paper presents an ultrasensitive single-axis in-plane micro-optical-electro-mechanical-system (MOEMS) accelerometer based on Talbot effect of dual-layer gratings. The MOEMS accelerometer achieved the acceleration sensitivity of 0.74 V/g (with better than 0.4% nonlinearity), a bias stability of 75 μg, and a resolution of 2.0 mg. However, to me, the innovation is moderate, especially when compared with their previous work, making it hard to give a higher rate. Overall, there are a number of issues that should be addressed before publication:
1. The authors claim that the mechanical structure of the accelerometer is investigated and optimized, while the optimization method and process should be given in detail. To be honest, I did not see sufficient new methods judging from its sensing methodology and reported design.
2. The authors should clearly clarify the novelty of this work, especially when compared with their previous work. Say, the acceleration sensitivity of this work is 0.74 V/g, while the result of the previous work can reach up to 3.1 V/g.
3. All of the figures have low resolution and should be redrawn.
4. Figure 2 shows the optical displacement sensitivity of 20%/μm, which is not a very satisfactory value.
5. I am interested in the noise suppression effect in terms of phase modulation and demodulation. The authors are encouraged to give more details.
6. There are some typos and errors, and the language should be polished.
